# Surface Modification Effect and Electrochemical Performance of LiOH-High Surface Activated Carbon as a Cathode Material in EDLC

**DOI:** 10.3390/molecules26082187

**Published:** 2021-04-10

**Authors:** Zambaga Otgonbayar, Sunhye Yang, Ick-Jun Kim, Won-Chun Oh

**Affiliations:** 1Department of Advanced Materials Science & Engineering, Hanseo University, Seosan-si, Chungnam 356-706, Korea; zambagaotgonbayar@gmail.com; 2Korea Electrotechnology Reserch Institute, 12, Boolmosan-ro, 10beon-gil, Seongsan-gu, Changwon-si, Gyeongsangnam-do 51543, Korea; shyang@keri.re.kr (S.Y.); ijkim@keri.re.kr (I.-J.K.); 3Department of Chemical Engineering, Inha University, 100 Inha-ro, Michuhol-gu, Incheon 22212, Korea; 4Anhui International Joint Research Center for Nano Carbon-based Materials and Environmental Health, College of Materials Science and Engineering, Anhui University of Science & Technology, Huainan 232001, China

**Keywords:** LiOH-treatment, high surface area of activated carbon, cathode material, electrochemical performance

## Abstract

This study aimed to improve the performance of the activated carbon-based cathode by increasing the Li content and to analyze the effect of the combination of carbon and oxidizing agent. The crystal structure and chemical structure phase of Li-high surface area activated carbon material (Li-HSAC) was analyzed by X-ray diffraction (XRD) and Raman spectroscopy, the surface state and quantitative element by scanning electron microscopy with energy dispersive X-ray spectroscopy (SEM-EDX) and the surface properties with pore-size distribution by Brunauer–Emmett–Teller (BET), Barrett–Joyner–Halenda (BJH) and t-plot methods. The specific surface area of the Li-YP80F is 1063.2 m^2^/g, micropore volume value is 0.511 cm^3^/g and mesopore volume is 0.143 cm^3^/g, and these all values are higher than other LiOH-treated carbon. The surface functional group was analyzed by a Boehm titration, and the higher number of acidic groups compared to the target facilitated the improved electrolyte permeability, reduced the interface resistance and increased the electrochemical properties of the cathode. The oxidizing agent of LiOH treated high surface area of activated carbon was used for the cathode material for EDLC (electric double layer capacitor) to determine its electrochemical properties and the as-prepared electrode retained excellent performance after 10 cycles and 100 cycles. The anodic and cathodic peak current value and peak segregation of Li-YP80F were better than those of the other two samples, due to the micropore-size and physical properties of the sample. The oxidation peak current value appeared at 0.0055 mA/cm^2^ current density and the reduction peak value at –0.0014 mA/cm^2^, when the Li-YP80F sample used to the Cu-foil surface. The redox peaks appeared at 0.0025 mA/cm^2^ and –0.0009 mA/cm^2^, in the case of using a Nickel foil, after 10 cycling test. The electrochemical stability of cathode materials was tested by 100 recycling tests. After 100 recycling tests, peak current drop decreased the peak profile became stable. The LiOH-treated high surface area of activated carbon had synergistically upgraded electrochemical activity and superior cycling stability that were demonstrated in EDLC.

## 1. Introduction

Activated carbon (AC) has specific properties, such as high specific area, balanced conductivity, high volume of porous structure and high stability in acidic condition (pH ≤ 7), that potentially make it an ideal material for the half-cell electrode [1,2,3,4,5]. Graphitic material is mostly used as a high-efficient electrode-material. The properties of high surface area activated carbon (HSAC) can be further re-shaped to instigate specific advantageous activation properties [6,7,8]. However, the elasticity of the carbon surface hinders absorption of the electrolyte into the electrode and weakens the output properties of the hybrid electrode due to the poor wettability on the carbon surface. Therefore, it is important to improve the hydrophilicity and output properties of the hybrid electrode through surface modification containing acidic functional-groups to meliorate the wetting properties of the water [9]. The surface is generally modified using chemical reagents such as KMnO_4_, H_2_O_2_, NaOH and LiOH [10,11,12] due to the various specifications for the activity of AC [13,14,15]. Alkaline hydroxide can be used as an activating agent to create a porous structure at high temperature [16].

The creation of the porous structure and surface modification is the most widely used method of improving the electrochemical properties of the activated carbon (AC). If an AC with an enormous surface area is used as a substrate and is reacted with lithium hydroxide (LiOH) or potassium hydroxide (KOH) under high-temperature conditions, an HSAC with a large micropore structure may be obtained. In order to increase the number of hydrophilic groups or change the activated carbon surface to hydrophilicity, acidic treatment and oxidizing agents capable of reducing the number of basic functional groups are used. In addition, a hydrophilic metal oxide (transition metal-based semiconducting oxide) is introduced to increase the wettability, electron transfer, and ion transfer effects of the hybrid electrode. Surface hydrophilicity control technology requires advanced carbon technology and evaluation technology. Developing these technologies quickly will require a batch-type hydrothermal synthesis method and a raw material control technology by microwave method that can support fast mass production [17,18]. The improvement of the output and long-term reliability of the hybrid capacitor depends on the surface control technology of AC and the conductive material.

The following sub-groups are important goals of this research: (a) surface hydrophilicity control technology improves the output characteristics by improving the electrolyte impregnation at the interface of the electrode and activated carbon and reducing the interface resistance, (b) improves the long-term reliability of the cell by inhibiting the corrosion of the hydrogen storage material of the cathode through the hydrophilicity control technology and catalytic function technology of the activated carbon surface, and (c) batch-type hydrothermal synthesis and microwave control technology for mass production.

In this study, LiOH- treated HSACs were successfully synthesized with surface modification that can activate the ion transfer and electrolyte impregnation. The phase structure, surface morphology, element analysis, functional groups, and electro-chemical properties of the modified (including surface coating and surface structure modification) activated carbon were analysed using XRD, SEM, EDX, FTIR, BET and a Potentiastat apparatus. LiOH-treated carbon material used for the working electrode. As-prepared Li-YP80F sample proved to have exceptional electrochemical performance, with high pore distribution and specific surface area. A LiOH-treated HSAC had good stability under a 10 and 100-cycling test. The peak current density and peak separation profile was good in Li-YP80F sample. A Li-YP80F electrode had good stability under a 100-cycle test. By comparing these three cycling performance curves, it can be seen that there is a process of activating capacity that happens during the first cycles. This approach presented herein offers a promising route for the rational design of a new class of supercapacitors.

## 2. Results

### 2.1. Crystallography and Morphology Analysis

The XRD pattern of LiOH-treated high surface area of activated carbon after heat treatment at 600 °C is shown in Figure 1. Figure 1a shows the XRD-pattern of Li-graphite, Li-YP50F and Li-YP80F. In all three samples, the clearest peak corresponded to the XRD pattern of Li_2_CO_3_. The diffraction peaks at 20 degrees of 21.46°, 29.52°, 30.63°, 32.0°, 34.22°, 36.95°, and 59.91° equated to the (110) (111) (202) (002) (112) (311) and (204) crystal planes of Li_2_CO_3_ (JCPDS. 22-1141). XRD peaks associated with Li_2_O, LiC and Li-composite were also observed, possibly due to the side reaction between the oxidizing agent and carbon material. These side peaks belonged to the XRD pattern of Li_2_O (JCPDS. 09-0355), LiC (JCPDS. 14-0649) and Li (JCPDS.15-1401). 

The XRD pattern of Li-Graphite exhibited one sharp peak at 26.78°, which is a member of graphite and lithium carbide. XRD graphs of the pure YP50F and YP80F are displayed in Figure 1b. Comparison of these results revealed the lithium-carbon nature of the Li-HSAC samples, which confirmed the successful synthesis of the final sample.

The surface state and morphology analysis of the samples are displayed in Figure 2. The LiOH treatment with high-temperature calcination completely changed the surface structure of the samples. The oxidizing agent was irregularly agglomerated and successfully up-loaded on the surface of the carbon materials due to the surface functional group. The quantitative analysis of the main element was analyzed by an EDX measurement incorporated by SEM.

Figure 3 shows the presence of the main elements of the samples. The C element is the principal element of the carbon and the O is the main element of the oxidizing agent. The main element of the oxidizing agent was obtained. This result confirmed the atomic weight of the main elements in the final samples. The crystallinity and SEM analyses of the agent’s distribution confirmed that it was successfully up-loaded. The results indicated that the re-structuring may modify the performance of the bare high surface area of activated carbon.

### 2.2. The Functional Group and Surface Area Analysis

The study of how the surface and functional group of a carbon material changes after LiOH treatment and calcification will determine how the material interacts with alkaline electrolyte ions. The surface functional group and surface area with pore distribution of the samples were analyzed by a Boehm titration and N_2_-adsorption-desoprtion isotherm. Boehm titration is an acid-base titration method that is used to determine the number of surface oxygen groups (acidic or basic) present on LiOH-treated carbon surfaces. This method is usually used complementary with other methods such as Fourier Transform Infrared Spectroscopy (FTIR).

In the titration, three different sodium-based solutions were used to neutralize the functional groups. Specifically, NaOH neutralizes the Bronsted acid group, NaHCO_3_ and Na_2_CO_3_ neutralize the carboxylic acid and lactonic groups, and this neutralization activity depends on the pKa of each solution [19]. To analyze the quantitative of functional group, Equations (8) and (9) were used and the data are summarized in Table 1. According to the titration results, the values of the surface functional groups were greater than the purpose of this study, which demonstrated that our LiOH-treatment experiments provided a proportionate number of acidic functional groups on the carbon surface and reduced the radical functional-groups, as shown in Figure 4. Depending on the ratio of acidic and basic-groups, the electrolyte impregnation at the interface of the working electrode can be improved, and the interface resistance can be reduced. The surface chemistry of the high surface area of AC strongly affects the electrochemical measurement. According to previous research, the surface functional group on porous-structured carbon material can achieve a high charge-storage and electrochemical performance [20,21,22,23]. In addition, the number of functional groups was increased by nitric acid treatment, which enhanced the wettability and increased the capacitance in acid and base-electrolyte solutions [24]. Furthermore, our analysis results confirmed that Li-HSAC had numerous acidic functional groups which can favorably act-on the diffusion and transport of the alkaline electrolyte ions.

The chemical bond vibration and the functional group were analyzed by FTIR, which is based on an attenuated total-reflection (ATR) method. Figure 5 shows the FTIR spectrum of the samples, which consisted of three main peaks. The main peaks of the carbon material were observed at around 1400 and 3000 cm^−1^ wavenumber regions. The presence of the lithium peak was observed at around 850 cm^−1^ region. The peak face of Li-graphite was different because the lithium peak was not observed. The pore-size, volume and specific surface area were analyzed by an N_2_-adsorption-desorption isotherm.

Figure 6 shows the nitrogen adsorption–desorption isotherms of Li-YP50F, Li-YP80F and graphite. The electrochemical properties of nanocomposites were correlated with the BJH and BET analysis results. The relative pressure of the analysis was 0.4–1.0 P/P_0_, and the relevance between pore-division and isotherm classified to the H1-type. The H1-type identifies the narrow pore-size state of the material. HSAC with large micropore structure may be obtained due to the LiOH agent and high temperature calcination. From BET analysis, the total pore volume and mean pore diameter of raw carbon material is reduced due to the oxidizing agent treatment. LiOH-oxidizing agent gathered in the disordered pore structure of the YP50F and YP80F. In the case of the graphite, the LiOH located on the surface. The pore distribution of the LiOH-treated samples was analyzed by BJH and T-plot method. According from the result, mesopore and micropore volumes and pore surface area were decreased. The micropore volume and surface area of the YP80F was high, the high amount of LiOH-oxidizing agent can inter-collected in the pore. The inter-collected amount of the LiOH can positively improve the Li+ ion transition between cathode and anode of EDLC, then it can upgrade the performance of the EDLC material.

The results of Li-HSAC pore-volume and specific surface area are summarized in Table 2. According to the summary results of BET and BJH, the surface area and total pore mass (mesopore and micropore) of the graphite was increased after the LiOH treatment, whereas the values were decreased for the bare-YP50F and YP80F. In the calculation of the t-plot, the total surface area of Li-YP50F was 43.5 m^2^/g and of Li-YP80F was 1063.2 m^2^/g. The micropore volume of Li-YP80F decreased from 1.084 to 0.511 cm^3^/g, while the mesopore volume increased from 0.100 to 0.143 cm^3^/g. The BET and BJH analysis results showed a similar decrease in the two main samples. The mesopore size and specific surface area were noticeably reduced after LiOH-treatment and calcination process. The micro-sized pore state and high surface area are the main parameters that are valuable for framing ion-transport tunnels in electro chemical reactions. 

Furthermore, the surface acidic group can improve the utilization rate of the electrode by facilitating the passage of ions into the micro-pores and improving the surface utilization rate of the pores. Based on these analysis results, the volume of the pore-distribution (micro, meso) reduced, which negatively affect the ion transport of the KOH electrolyte. However, the presence of the surface acidic group can enhance the ion-transport through the enhanced micropore structure of the Li-HSAC material, because the acidic group can improve the wettability and electronegativity, which in turn affect the electrolyte impregnation. In addition, the Li-YP80F sample had high surface area, which demonstrated the more active interfacial site between the alkaline electrolyte and the cathode. Previous studies have shown that the surface area and pore size are the main factors influencing the performance of the electrode material [25,26]. The micropore-structured carbon material showed a high capacitance with continuous charge and discharge due to the oxygen-containing functional group. The specific capacitance of the LiOH-treated graphite, YP50F and YP80F carbon were calculated, and all analysis results are summarized in Table 3. According to the analysis result, the Li-YP80F sample had a high value of specific capacitance and integration area. This sample had high number of micropore distribution and high surface area with hydrophilicity, and the porous (micro) structure with a high surface of carbon material is an ideal material for the electrode on the EDLC. All of the above factors are present in the Li-YP80F material, which confirmed that the Li-YP80F sample had high electrochemical properties.

The wettability of the LiOH-treated carbon materials was tested by gravimetric methods using water and 5% KOH (Figure 7). The measurement-based of determining how much amount of solvent (water, 5% KOH) absorbed in the working electrode which using a LiOH-treated carbon material and pure carbon material. The results of the wettability properties of all samples are summarized in Bit-graph. According from the summarized result, LiOH-treated sample has high wettability properties, due to the acidic groups of the surface which can affect the impregnation of the electrolyte. The main purpose of the study was to change the surface of the carbon material by the method of functional groups with hydrophilic and hydrophobic properties, depending on the type of electrolyte applied. The results of the experiments confirm that the surface of the carbon material has been successfully modified to a hydrophilic quality using LiOH-oxidizing agents.

Figure 8 expresses the Raman spectra of the LiOH-treated graphite, YP50F and YP80F samples. Raman spectroscopy is the chemical-analysis method which can analyze the chemical structure phase and molecular interaction of matter. The principle of Raman spectroscopy is a light-scattering technique, where the light interact to the molecule bond and the result provides the information of carbon material (D and G-band). The assignment of the D and G peaks is straightforward in the “molecular” picture of carbon materials. The Raman spectrum of LiOH-YP50F and LiOH-YP80F is dominated by two prominent features. The G band near 1597 and 1591 cm^–1^ is due to the degenerate zone-center (G) optical phonon mode with E_2g_ symmetry, which is found in most graphitic materials. The most important is the D band near 1332 and 1338 cm^–1^ due to a one-phonon scattering involving a phonon near the K point and scattering due to defects [27].

Two broad peaks at 2657, 2662, 2914 and 2904 cm^–1^ observed in the spectrum of LiOH was also detected in the Raman spectrum of LiOH-treated YP50F and YP80F. These two peaks are assigned to the asymmetric (B_g_) and symmetric (A_g_) stretches [28]. The five peaks are dominated in LiOH-Graphite sample. The strong band at 2693 cm^–1^ is due to a two-phonon scattering involving $A_{1}^{\prime }$ phonons near the K point and is called the 2D band. The 2D band is usually stronger than the G* and G’ band although it is due to a two-phonon scattering process. The triple resonance process, where the electron and the hole scatter simultaneously, can also explain the two-phonon scattering. The characteristic peak of LiOH not obtained in LiOH-Graphite sample, due to the peak intensity of each components.

### 2.3. Electrochemical Test

The high surface area of YP50F, YP80F activated carbon and graphite was modified with LiOH-oxidizing agent and heat treatment (600 °C). HSAC materials with a micro-porous structure can be synthesized by (LiOH) oxidizing agent treatment for the high surface area of activated carbon under the high temperature. In addition, after the activated carbon reacting with oxidizing agents, the number of basic functional groups decreases, the number of hydrophilic groups increases, or the hydrophilicity of the activated carbon surface improves, which can improve the long-term reliability of the cell.

The Li-HSAC samples were used for the working electrode by following a Doctor blade” method. The electrochemical performance and work-function of the electrode were analyzed by a cyclic voltammetry (CV) and chronoamperometry (CA) technique. The CA technique is useful that Potentiastat transient measurement is a favorable method to analyze the capacity-rate curve of the working electrode [29]. The CV tests were carried out in a three-electrode system. In this performance, Cu-foil and Ni-foil was used as a current-collector and the high ionic conductivity of the KOH basic solutions was used for the electrolyte [30]. KOH is a basic electrolytes and it has been the most extensively used because of its high ionic conductivity and it can support the cyclic stability of the working electrode.

The CV test was run for 10 and 100-cycles with 100 mV/s scan rate. The CV graph and current value were different in each sample according to the working electrode. The CV result of Li-YP80F is shown in Figure 9a,b. The anodic and cathodic peak current value and peak segregation of Li-YP80F were better than those of the other two samples, due to the micropore-size and physical properties of the sample. The oxidation peak current value appeared at 0.0055 mA/cm^2^ current density and the reduction peak value at −0.0014 mA/cm^2^, when the Li-YP80F sample used to the Cu-foil surface. The redox peaks appeared at 0.0025 mA/cm^2^ and −0.0009 mA/cm^2^, in the case of using a Nickel foil, as shown in Figure 9b.

As shown in Figure 9c, the CV curve had one sharp peak in the oxidation state. The oxidation peak appeared at 0.0035 mA/cm^2^ and the reduction-peak at the 0.00042 mA/cm^2^ regions. When the Li-YP50F covered on the nickel foil, the redox peaks appeared at 0.00231 mA/cm^2^ and −0.00051 mA/cm^2^ current density regions, as shown in Figure 9d. For the Li-graphite, the current density of the oxidation peak appeared at 0.0053 mA/cm^2^ and the reduction peak at (−0.00045 mA/cm^2^), as shown in Figure 9e. In the case of Li-graphite coated on the nickel foil surface, the oxidation peak appeared at 0.0013 mA/cm^2^ and reduction peak appeared at −0.00083 mA/cm^2^ region, as shown in Figure 9f.

The electrochemical properties of the Li-HSAC sample depend on three main factors: the functional group, surface area, and pore size distribution. The interfacial area, the ion-transfer between the electrode surface and the electrolyte depend on the pore size and the surface area. The analysis results revealed that the specific surface area and pore size were higher for Li-YP80F than for the other two samples, which strongly improved the electrochemical performance. 

The use of Li-YP80F, a material with a high surface area, as a working electrode allows for more charge exchange and, on the other hand, allows more electrons to accumulate over a given period of time and conduct more current. The charge exchange happens at a fixed rate per unit area at the surface of the electrode. Also, the numerous acidic functional groups of Li-YP80F enabled it to maintain the permeability of the electrolyte and reduce the resistivity between the electrolyte and working electrode. In the *CV* graph, the peak separation is dependent on the electron transfer kinetics and conductivity of the base aqueous electrolyte. In addition, the high conductivity of the as-prepared KOH electrolyte influenced the performance of the working electrode [31]. Specifically, the electron transfer of the working electrode was high when the working electrode was dipped in electrolyte. The conductivity of the base-electrolyte is conditional-on the smaller ionic hydrated-radius and rapid-motility of OH^−^ [32,33,34]. In addition, the *CV* graph profile was changed due to the carbon material of the base substrate, for example, the oxidation peak-current state changed from 0.0055 mA/cm^2^ to the current density state to 0.0035 mA/cm^2^. In addition, the Cu plate collector has a conductivity of 0.6 S/cm and it can support the activity of the base sample. The specific capacitance, area of the integration, energy density (Wh/kg) at given powder density (kW/kg) are important factors which define the performance of the working electrode. Aforementioned mentioned all factors were calculated by following the below Equations (1)–(7). All calculated results are summarized in Table 3. The calculation of specific capacitance needs to summarize some units, such as charge, capacitance.
(1)$$charge=\;\frac{integrated\;\;area\;of\;CV}{2xScan\;rate}$$
(2)$$capacitance=\;\frac{charge}{potential\;window}$$
(3)$$specific\;capacitance=\;\frac{capacitance}{mass\;of\;active\;material}$$
(4)$$\mathrm{Combined}\;\mathrm{equation}\;\mathrm{was}\;\mathrm{the}\;\mathrm{Cp}=\;\frac{A}{2mk\;\left(V2\;-V1\right)}$$
where Cp is the specific capacitance in F/g. *A* is the area inside the *CV* curve having units AV, *m* is the mass of active material, *k* is the scan rate of *CV* in volts per second and (*V*_2_ − *V*_1_) is the potential window of *CV*.

The calculation of energy density (Wh/kg) and powder density (W/g).
(5)$$C=\frac{i\;\times \;t}{m\times \Delta U}$$
(6)$$dE\;\left(\frac{Wh}{kg}\right)=0.5\times C\times {\left(\Delta U\right)}^{2}$$
(7)$$dP\;\left(\frac{kW}{kg}\right)=\left(\frac{dE}{1000}\right)÷\left(\frac{t}{3600}\right)=3.6\times \frac{dE}{t}$$
where *C* is the capacitance, Δ*U* is potential window, *t* is discharge time, the factor 3.6 is used as correcting the unit form gram and second into kilogram and hour.

The electrochemical stability of cathode materials was tested by 100 recycling tests. The peak-to-peak separation and high redox peaks appeared in the Cu-foil used three electrode system. After 100 recycling tests, peak current drop decreased the peak profile became stable. The electrochemical stability of cathode materials was tested by 100 recycling tests, as shown in Figure 10. The Cu-current collector of the working electrode was corroded, and the surface of the electrode did not change after the electrochemical test. In the case of nickel foil, there is no corrosion process appeared and the current peak did not significantly change. The corrosion process can be happened due to the electrolyte and concentration. 

Figure 11 shows the current value of the working electrode (vs. Ag/AgCl) according to time-condition in 5% KOH electrolyte. The current of the working electrode was high in the first cycle and decreased to a certain extent after 10 cycles. The profile of Li-YP80F had a uniform structure and this result confirmed that Li-YP80F sample had more stable structure and the working performance.

Figure 12 shows the CA analysis result of the bare-carbon material and modified Li-HSAC. The fundamental technique of the CA is to analyze the potential and capacity of the working electrode according to time, as is done in the GCD method. The analysis revealed a high current on Li-YP80F, which confirmed that the material had a high work capacity and was suitable for the cathode electrode in a half-cell. The flat band potential of Li-YP50F, Li-YP80F and graphite was analyzed by the Mott–Schottky method. Depending on the potential of the flat strip, the material type can be defined as n or p-type. As shown in Figure 13, the flat band potentials of the nanocomposite were 0.009 V, 0.074 V and −0.095 V, respectively. The Mott–Schottky slope graph shows that the Li-YP50F and Li-graphite are p-materials and the Li-YP80F is an n-type material. 

Figure 14 shows a schematic illustration of the Li-HSAC surface state and electrochemical analysis. The graphite had a layered-like structure with a hexagonal arrangement of carbon atoms, which provides sufficient area for locating the oxidizing agents. The Li-YP50F and Li-YP80F materials have different porous structures, and oxidizing agents can accumulate in the pores. The pore structure of Li-YP80F and Li-YP50F was sufficient for electrolyte-ion conduction and distribution, due to the hydrophilic properties and the high area of material interaction. These results confirmed that the Li-YP80F electrode had high capability and electrochemical property. The structure of the oxidizing agent can improve the interfacial area of the electrolyte and working electrode surface. Furthermore, the high pore size distribution of Li-YP80F afforded it with high ion-transfer performance. These electrochemical results confirmed that oxidizing agent treatment with calcination can enhance the performance of the high surface area of carbon materials. In addition, the values of the surface functional groups were higher than the purpose of this study, which demonstrated that our LiOH treatment experiments provided a sufficient number of acidic functional groups on the surface of carbon materials and reduced the basic-functional groups. Depending on the ratio of the acidic and basic groups, the electrolyte impregnation on the electrode surface and interface resistance can be reduced. Furthermore, our mentioned method and characterization proved that oxidizing agent (LiOH) treatment with a heating process can ameliorated the cathode electrode performance.

Recently, various activated carbon-based electrode materials have been used for electrochemical analysis, and the research has suggested the properties of surface area, pore size, and surface chemistry. These properties strongly influence the electrode capacity and charging/discharging performance used in the long-term cycle analysis [35,36,37]. The micro-sized pore and low specific surface area strongly affect the interaction site and block the ion-transmittance. In conclusion, the pore volume and surface area were significantly reduced on the HSAC after the LiOH treatment with the thermal process. However, the high total surface area of Li-YP80F enhanced the inter-facial site and reaction field between the electrolyte and electrode. The number of the acidic functional groups is greater than the target value, which supports the process of electrolyte ion transfer and diffusion through the micro-sized pores. The wettability condition of the electrode surface and low resistance with high charge transfer depend on the acidic functional group of the high surface area of activated carbon. These aforementioned factors combined to synergistically upgrade the electrochemical activity of the Li-HSAC nanomaterial.

## 3. Materials and Methods

### 3.1. Chemical Reagents

The YP50F and YP80F activated carbons were purchased from Kuraray (Tokyo, Japan). And the natural graphite was used in this experiment. The HSAC material was prepared by ball-milling at 350 rpm for 10 h. Lithium hydroxide (LiOH, 98%), sodium hydroxide (NaOH, 97%), sodium carbonate (Na_2_CO_3_, 99.9%), sodium bicarbonate (NaHCO_3_, 99.7%) and hydrochloric acid (HCl, 36 wt%) were purchased from Sigma Aldrich (Munich, Germany). Potassium hydroxide (KOH, 98%) were purchased from Samchun Pure Chemical Co., LTD (Gyeonggi-do, Korea).

### 3.2. Preparation of Hydrophilic Carbon

Into a prepared LiOH solution (250 mL) was added 20 g of raw HSAC material. The mix was stirred continuously for 5 h in room condition and then dried at 100 °C in an oven to form a powder. The LiOH-treated carbon samples were calcined at 600 °C for 2 h in the furnace to purify them and revamp their surface-state. All prepared high-surface-area carbon was grouped into one group, Li-HSAC.

### 3.3. Characterization

The crystal phases of the samples samples were examined between 20 = 10 to 70 at a scan rate of 1 min^−1^ using an X-ray diffraction instrument (SHIMADZU XRD-6000, Nakagyo, Kyoto, Japan) equipped with a Cu Ka X-ray source (1.5406 Å). The surface morphology state of the sample was analyzed by scanning electron microscopy with energy dispersive X-ray spectroscopy (SEM-EDX; JSM-5600 JEOL, Akishima, Tokyo, Japan). The pore volume and surface area were evaluated by the Barrett–Joyner–Halenda (BJH) and Brunauer–Emmett–Teller (BET) methods, respectively, and the whole experiment was conducted on a Micro Active for ASAP 2460 (Norcross, GA 30093, USA). The functional group and chemical bonds were analyzed by using a Fourier-transform infrared spectrometer (FTIR iS5, Thermoscience, Waltham, MA, USA), and the spectral resolution was 3.8 cm^–1^ background scanning speed and sample scanning speed are 20 scans, respectively. A Raman spectroscopy was performed using a Confocal-Raman imaging system with a 532.13 nm excitation laser (Renishaw in Via Reflex, NRS-5100, Easton, MD 21601, USA).

### 3.4. Boehm Titration

The surface functional groups of the Li-HSAC nanocomposites were analyzed by a Boehm titration method. A certain mass of sample was added to 0.05M of three different sodium-based solutions (NaOH, NaHCO_3_, Na_2_CO_3_). The mixture was shaken for 24 h at room temperature. The aliquot was withdrawn by a syringe centrifuge tube and rapidly rotated the sample (10,000 ppm/15 min), and 10 mL solution was taken by pipette and acidified by adding 0.05 M of HCl solution. In detail, 0.05M of 20 mL HCl was added into 10 mL of NaOH solution, 30 mL 0.05M of HCl was mixed with Na_2_CO_3_ aliquot and 20 mL HCl solution was added to the NaHCO_3_ aliquot solution. A 2/3 drop of 0.5% phenolphthalein indicator solution was added into each solutions and centrifuged for ¼ h. In the following back-titration process, 0.05 M NaOH solution was put-in slowly into the acid-treated mix until the color of the solution turned pink. The titration processes were carried out under ambient temperature conditions. The acidic groups were quantified by Equations (8) and (9) [18]:(8)$$\left[\mathrm{HCl}\right]V_{\mathrm{HCl}}=\left[\mathrm{NaOH}\right]V_{\mathrm{NaOH}}+\left(\frac{\eta_{\mathrm{HCl}}}{\eta_{B}}\left[B\right]V_{B}-\eta_{\mathrm{CSF}}\right)\frac{V_{a}}{V_{B}}$$
(9)$$\eta_{\mathrm{CSF}}=\frac{\eta_{\mathrm{HCl}}}{\eta_{B}}\left[B\right]V_{B}-\left(\left[\mathrm{HCl}\right]V_{\mathrm{HCl}}-\left[\mathrm{NaOH}\right]V_{\mathrm{NaOH}}\right)\frac{V_{b}}{V_{a}}$$
where [B] and V_B_ are the concentration and volume of the reaction base mixed with the carbon, providing the number of moles of reaction base that was available to the carbon surface for reaction with the surface functionalities. η_CSF_ denotes the moles of carbon surface functionalities on the surface of the carbon that reacted with the base during the mixing step. V_a_ is the volume of the aliquot taken from the V_B_, and [HCl] and V_HCl_ are the concentration and volume of the acid added to the aliquot taken from the original sample.

### 3.5. Cyclic Voltammetry (CV) Test

The electro-chemical properties of the sample were examined by a cyclic voltammetry (CV) test. The analysis was conducted on a PGP201 Potentiastat (A41A009) by using a three-electrode system. The CV test was recorded with a 100 mV·s^−1^ scan rate for 10 cycles. The applied potential range was from (−300 mV) to (200 mV) and the current value range was (1 A) to (−1 A). A platinum wire and Ag/AgCl were used as a counter and reference electrode. Li-HSAC material as the working electrode. The working electrode was prepared by following a “Doctor blade” method. Ethyl cellulose was used as a binding material and mixed with Li-HSAC in a 1:3 ratio. Then, a few drops of pure ethanol were added, and the resulting mixture was ground and used to veneer the copper foil-top. The size of the copper foil and nickel foil is 2 × 2 cm^2^. The Cu-foil current collector enhanced the capacity of the sample; this is due to the conductivity of copper being about 0.6 S/cm. The uniformity and thickness were metered by adjusting gap between the blade and the substrate. The main principal process of doctor-blading uses a frame with a reservoir coating liquid which is moving relatively to the substrate. When a constant movement between the blade and substrate, the semiliquid mixture spread onto the substrate to make a thin thin-sheet and after drying process it can turn into a gel-layer.

The typical three-electrode assembly was immersed in 5% KOH supporting electrolyte solutions. KOH is a basic electrolytes and it has been the most extensively used because of its high ionic conductivity and it can support the cyclic stability of the working electrode.

## 4. Conclusions

In summary, the working electrode was prepared by a doctor blade method using a LiOH-treated activated carbon with high surface area. The effect of this modification was analyzed by XRD, Raman, SEM-EDX, BET and BJH methods. From the results of material structure analysis, we attributed the unique structure of this material to the connection between an alkali metal and the high surface area of the activated carbon. The surface functional group was analyzed by a Boehm titration-method, which confirmed that the alkali metal-treated carbon material had numerous acidic groups above the target value. The acidic functional group of Li-HSAC enhanced the electrolyte ion-migration through the micro-sized pore structure. The evaluation of the electrochemical properties of Li-HSAC demonstrated that the high electrochemical properties were attributable to the total specific surface area, pore-size distribution, wettability condition and surface modification with an oxidizing agent (LiOH).

## Figures and Tables

**Figure 1 molecules-26-02187-f001:**
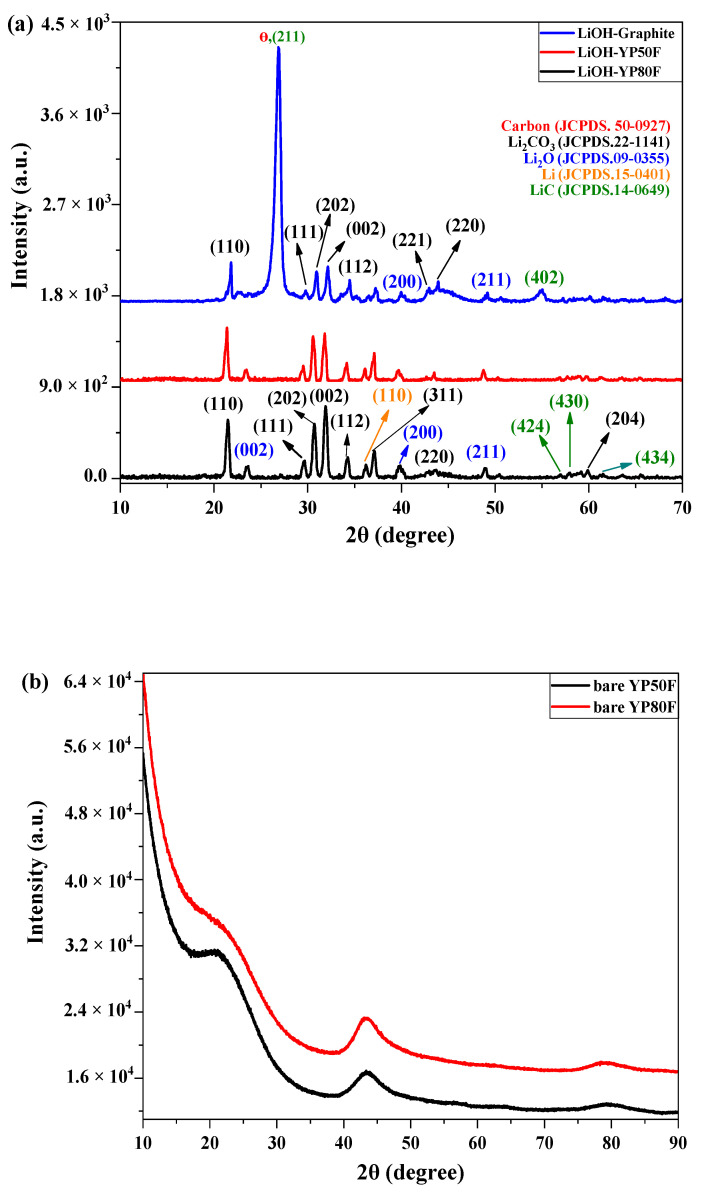
XRD result of (**a**) LiOH-treated YP80F, YP50F and graphite, (**b**) bare YP80F and YP50F.

**Figure 2 molecules-26-02187-f002:**
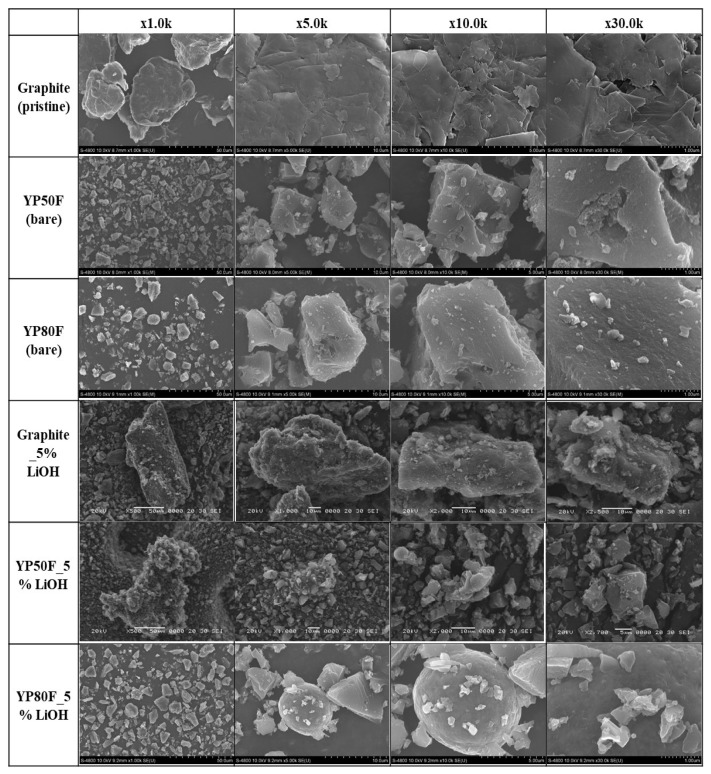
SEM images of bare graphite (pristine), YP50F, YP80F, graphite–LiOH, YP50F-LiOH and YP80F-LiOH.

**Figure 3 molecules-26-02187-f003:**
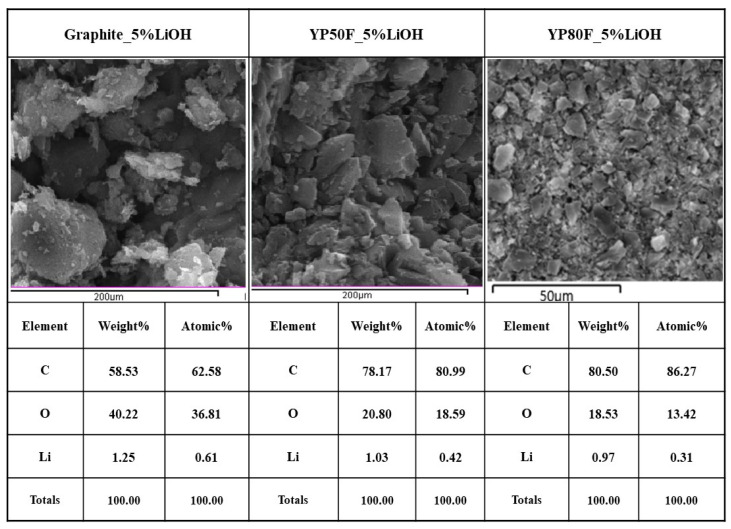
EDAX analysis of LiOH-treated YP50F, YP80F and graphite.

**Figure 4 molecules-26-02187-f004:**
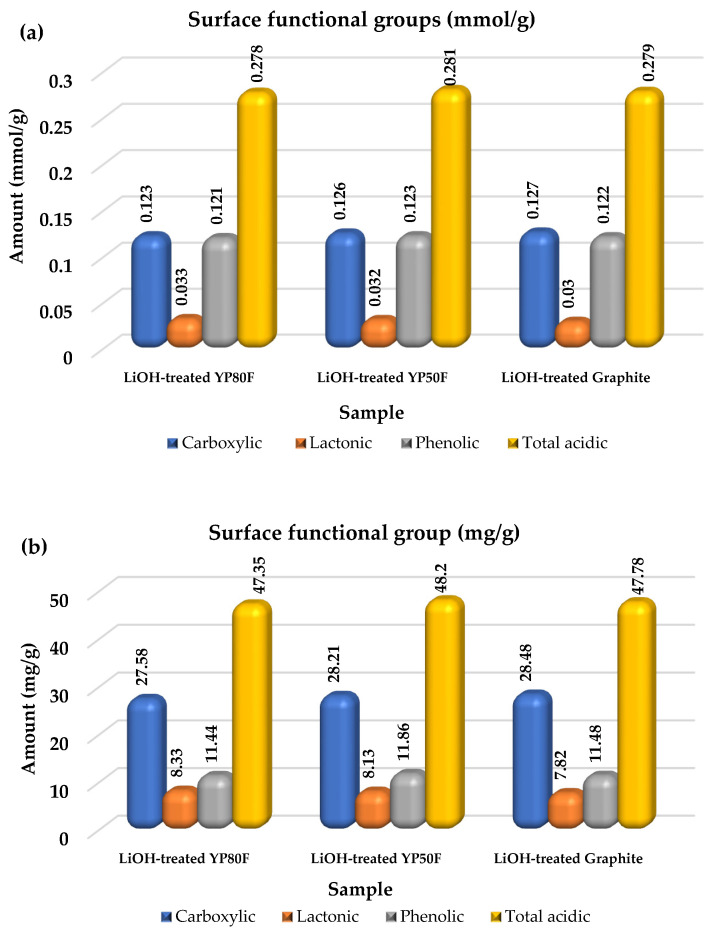
Boehm titration result of LiOH-treated YP80F, YP50F and graphite. (**a**) mmol/g unit, and (**b**) mg/g unit.

**Figure 5 molecules-26-02187-f005:**
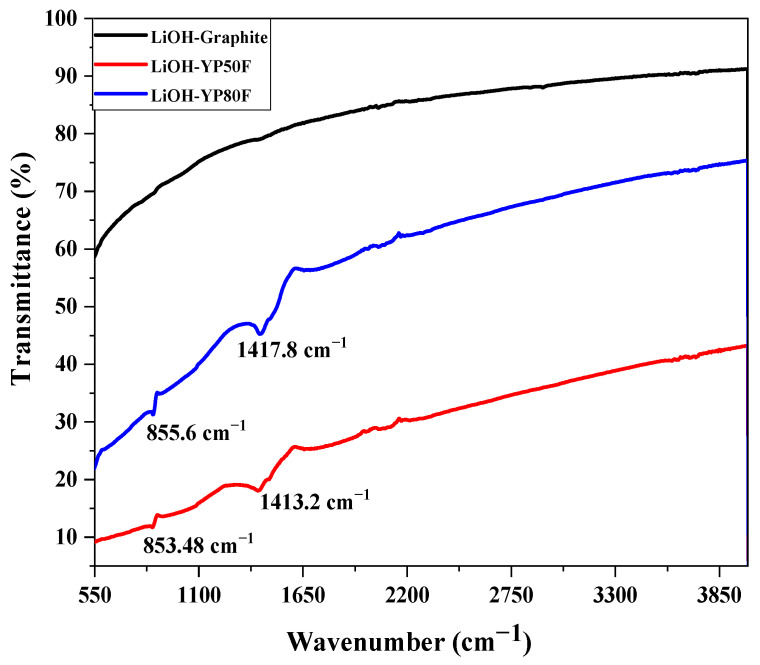
FTIR spectra of LiOH-treated YP80F, YP50F and graphite.

**Figure 6 molecules-26-02187-f006:**
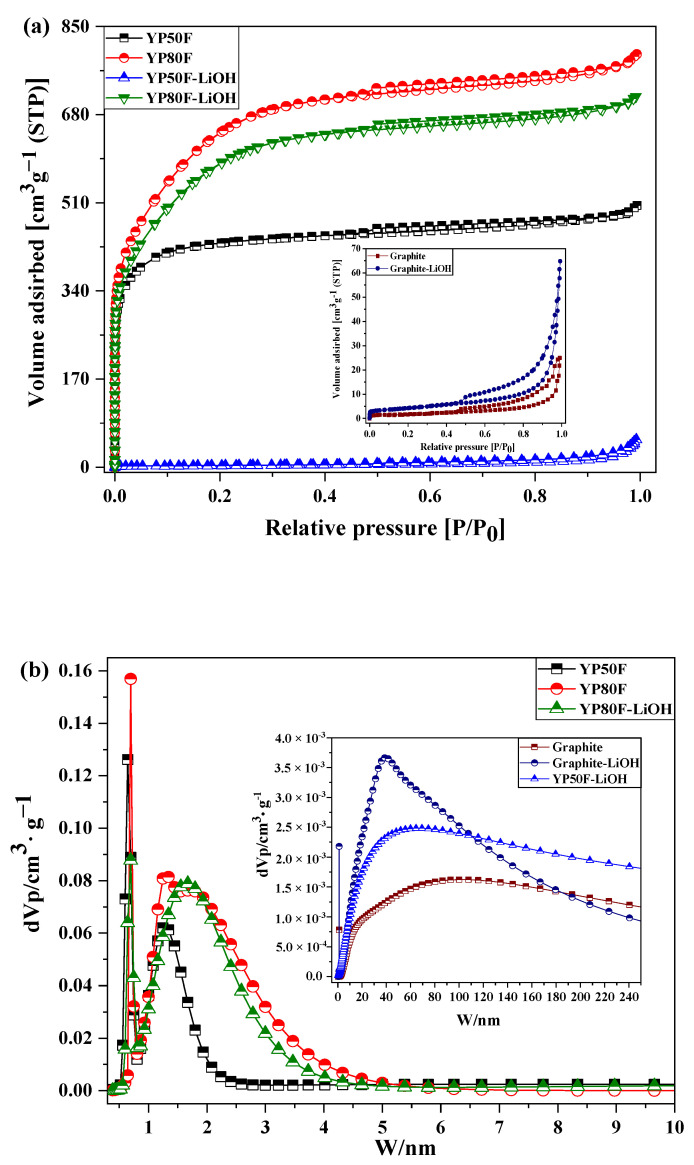
(**a**) Nitrogen adsorption–desorption isotherms, (**b**) NLDFT/GCMC Pore size distribution analysis, (**c**) BJH plot and (**d**) T-curve of graphite, YP50F, YP80F and LiOH-treated carbon materials.

**Figure 7 molecules-26-02187-f007:**
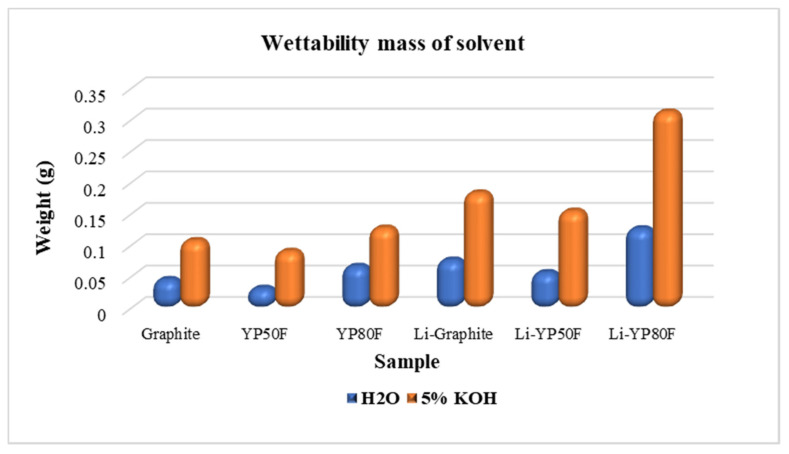
Bit-graph. The wettability of the carbon samples.

**Figure 8 molecules-26-02187-f008:**
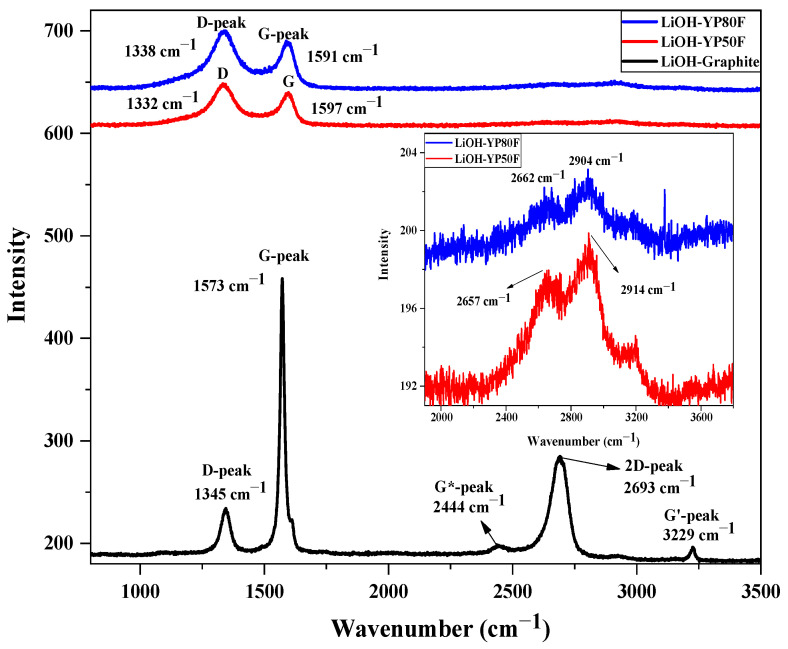
Raman spectra of the LiOH-Graphite, LiOH-YP50F and LiOH-YP80F sample (G*-two-phonon processes).

**Figure 9 molecules-26-02187-f009:**
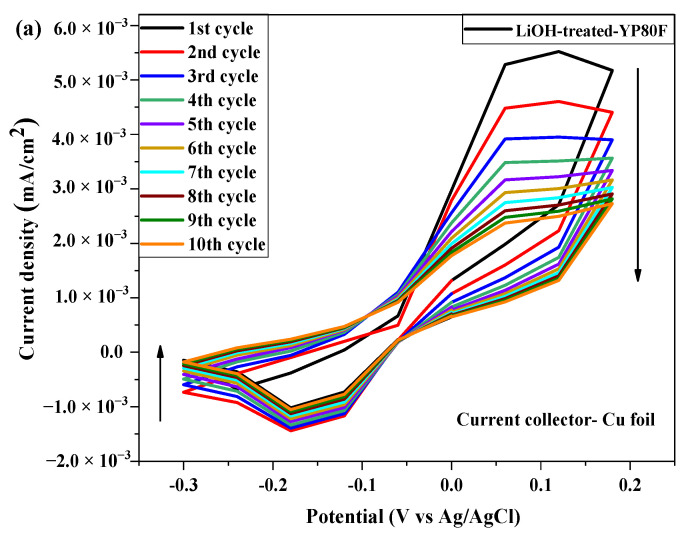
Cyclic voltammetry (CV) test of (current collector-Cu foil and Ni foil and 5% KOH electrolyte) under 10 cycles (**a**) LiOH-YP80F with Cu-foil; (**b**) LiOH-YP80F with Ni-foil; (**c**) LiOH-YP50F with Cu-foil; (**d**) LiOH-YP80F with Ni-foil; (**e**) LiOH-Graphite with Cu-foil; (**f**) LiOH-Graphite with Ni-foil.

**Figure 10 molecules-26-02187-f010:**
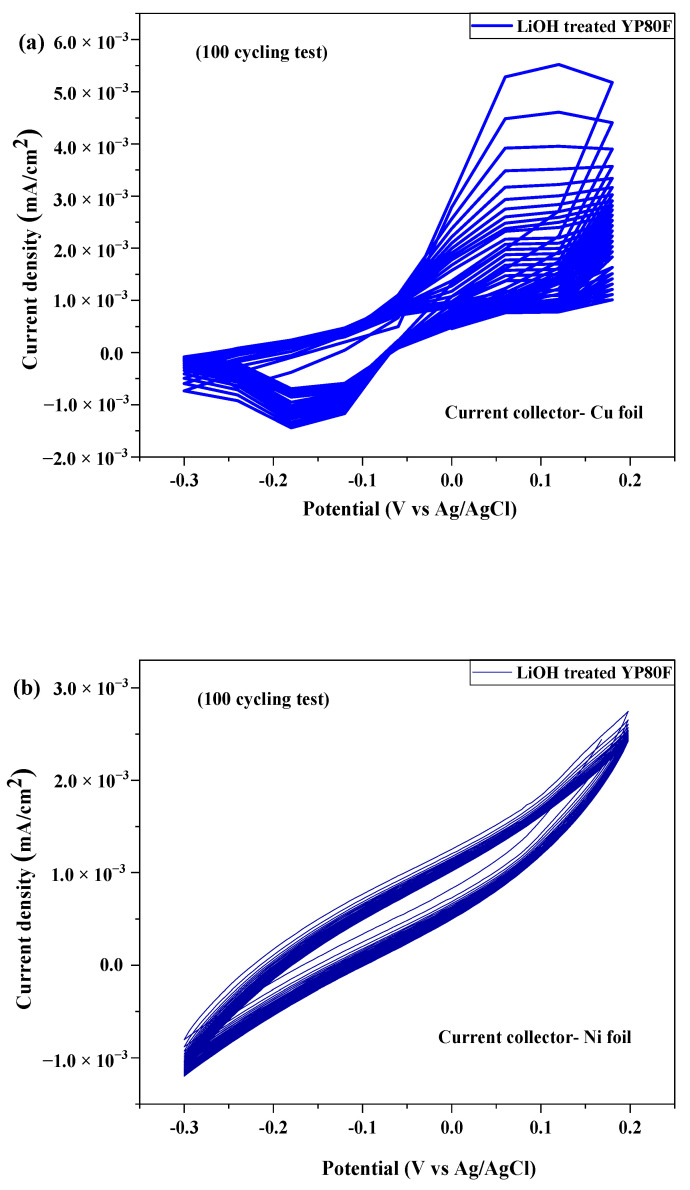
Cyclic voltammetry (CV) test (current collector—Cu foil and Ni foil and 5% KOH electrolyte) under 100 cycles. (**a**) LiOH-YP80F with Cu-foil; (**b**) LiOH-YP80F with Ni-foil; (**c**) LiOH-YP50F with Cu-foil; (**d**) LiOH-YP80F with Ni-foil; (**e**) LiOH-Graphite with Cu-foil; (**f**) LiOH-Graphite with Ni-foil.

**Figure 11 molecules-26-02187-f011:**
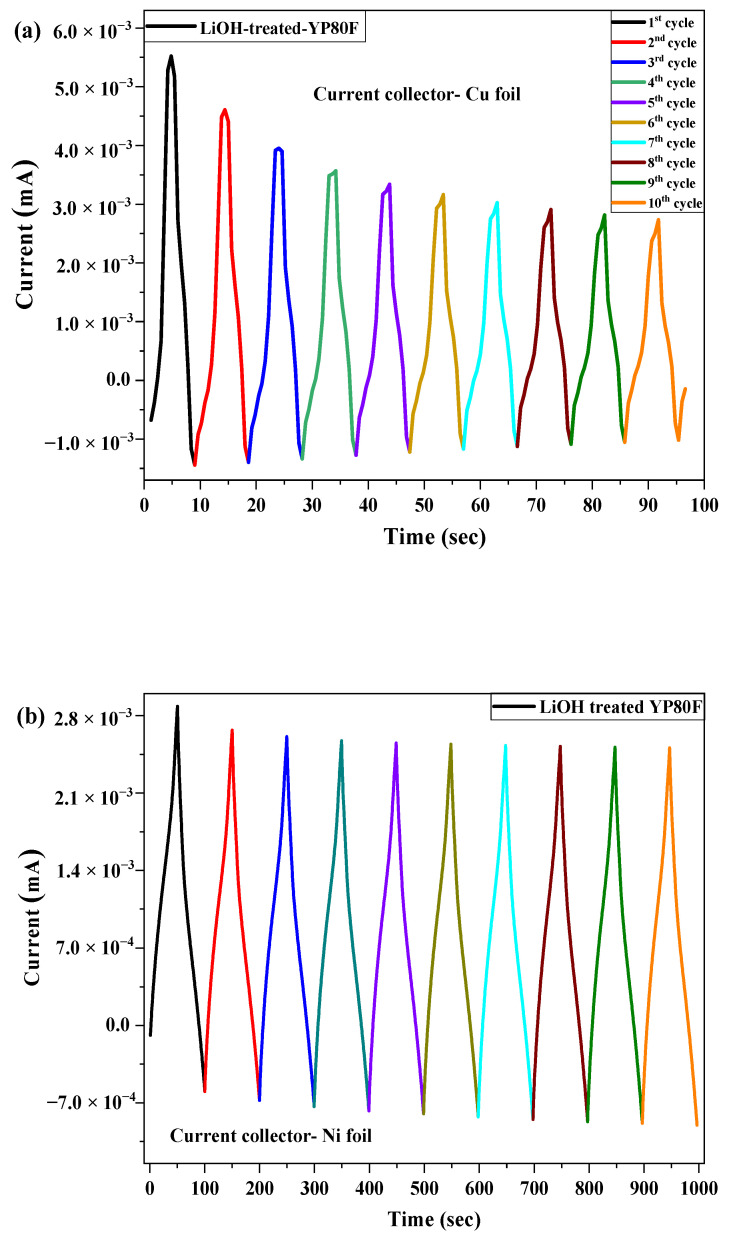
Stability test of LiOH-treated YP80F, YP50F and graphite (Cu-foil and Ni-foil current collector and 5% KOH electrolyte). (**a**) LiOH-YP80F with Cu-foil; (**b**) LiOH-YP80F with Ni-foil; (**c**) LiOH-YP50F with Cu-foil; (**d**) LiOH-YP50F with Ni-foil; (**e**) LiOH-Graphite with Cu-foil; (**f**) LiOH-Graphite with Ni-foil.

**Figure 12 molecules-26-02187-f012:**
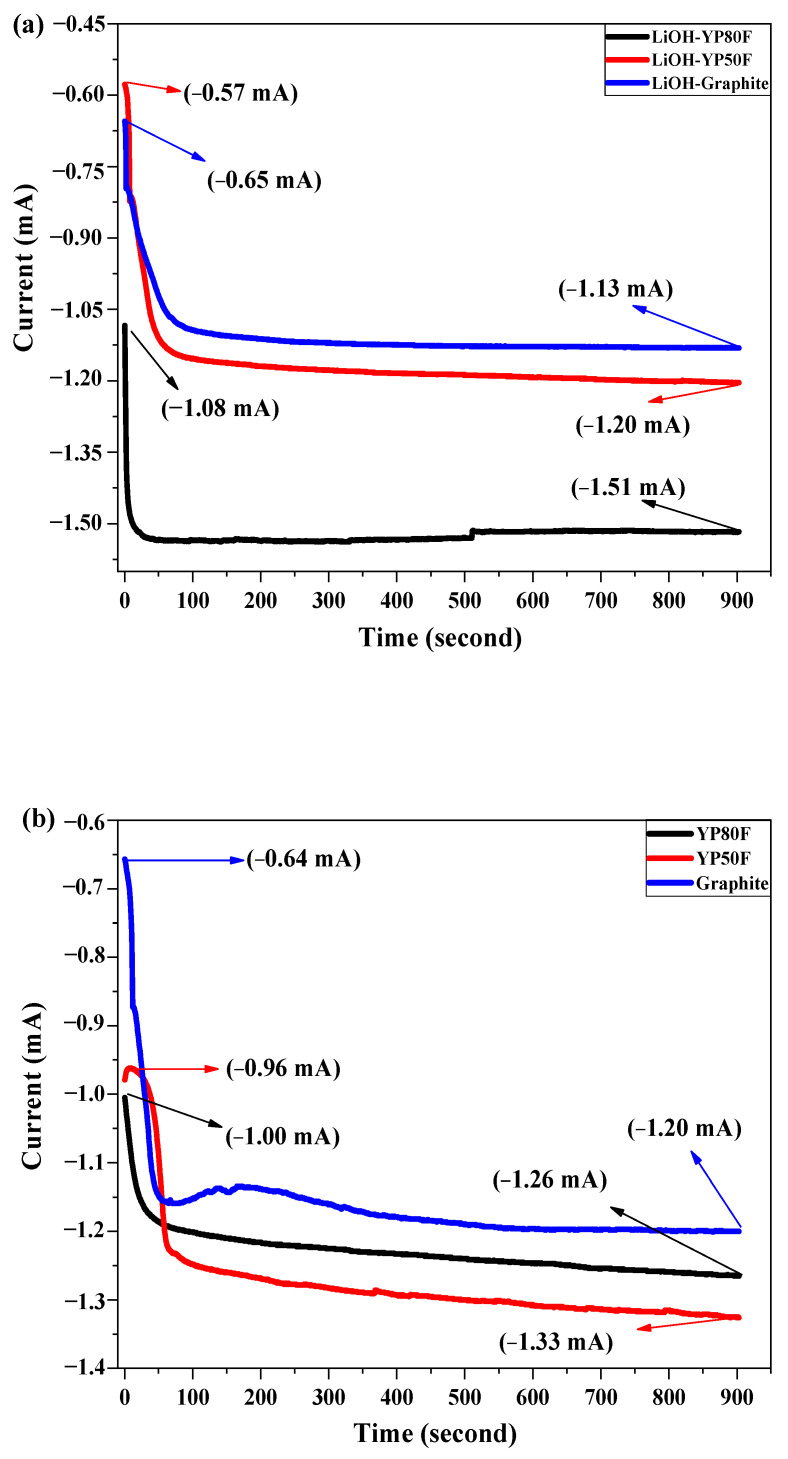
Chronoamperometry test of LiOH-treated YP80F, YP50F and Graphite (Cu-current collector and 5% KOH electrolyte). (**a**) LiOH-treated samples; (**b**) pure samples.

**Figure 13 molecules-26-02187-f013:**
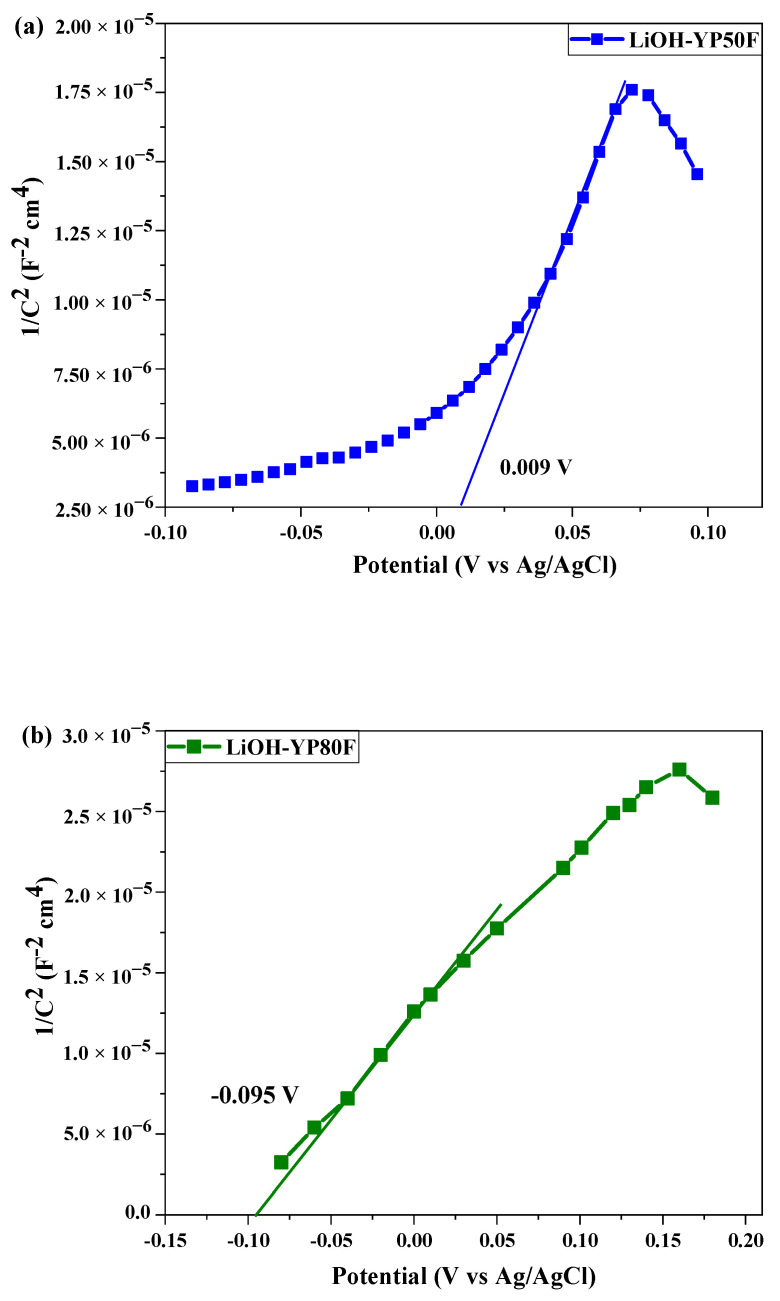
Mott–Schottky plot of LiOH-treated (**a**) YP50F, (**b**)YP80F, and (**c**) graphite (Cu-current collector and 5% KOH electrolyte).

**Figure 14 molecules-26-02187-f014:**
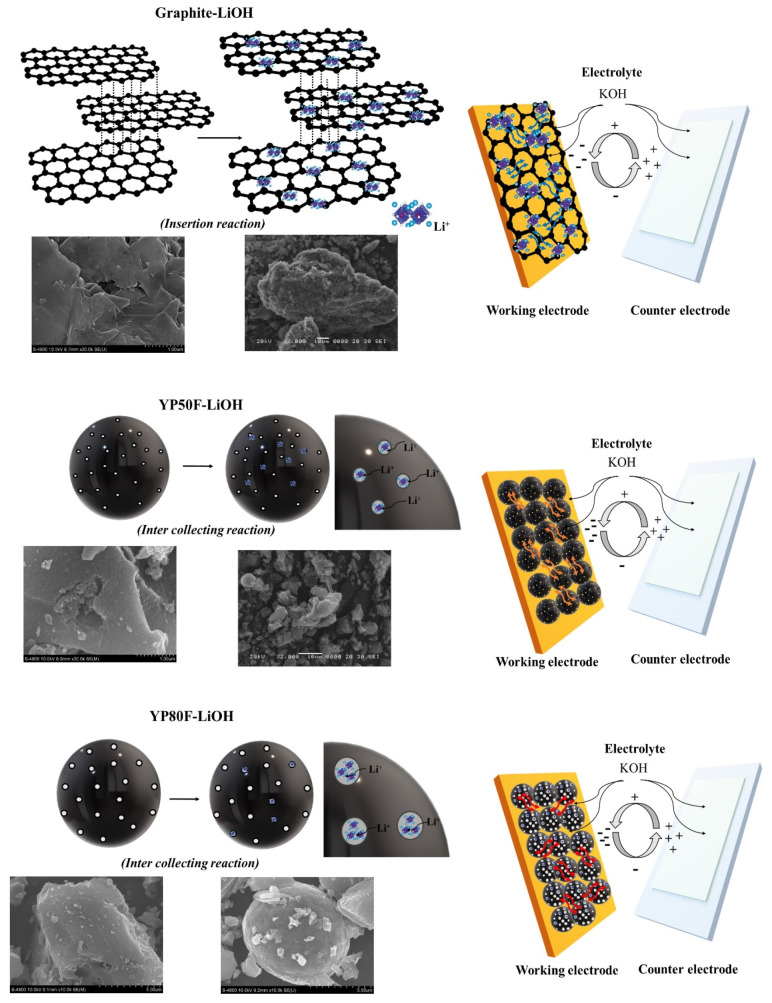
Schematic illustration of the preparation of LiOH-treated carbon material and electrochemical analysis.

**Table 1 molecules-26-02187-t001:** Surface functional group analysis of the LiOH-treated carbon materials.

		Surface Functional Group (mg/g)				Surface Functional Group (mmol/g)			
		Carboxylic	Lactonic	Phenolic	Total acidic	Carboxylic	Lactonic	Phenolic	Total Acidic
1	LiOH-treated YP80F	27.58	8.33	11.44	47.35	0.123	0.033	0.121	0.278
2	LiOH-treated Graphite	28.48	7.82	11.48	47.78	0.127	0.030	0.122	0.279
3	LiOH-treated YP50F	28.21	8.13	11.86	48.20	0.126	0.032	0.123	0.281

**Table 2 molecules-26-02187-t002:** Nitrogen adsorption–desorption isotherms of bare graphite, YP50F, YP80F and LiOH-treated samples.

**BET Analysis**						
**Samples**	**Surface Area (m^2^/g)**		**Total Pore Volume (cm^3^/g)**		**Mean Pore Diameter (nm)**	
Graphite	6.29		0.042		26.6	
YP50F	1676.1		0.776		1.9	
YP80F	2374.8		1.227		2.1	
Graphite-LiOH	15.2		0.094		24.6	
YP50F-LiOH	11.7		0.074		25.3	
YP80F-LiOH	2155.7		1.104		2.0	
**BJH Analysis**						
**Samples**	**Mesopore Pore Diameter (nm)**	**Micropore Surface Area (m^2^/g)**	**Mesopore Surface Area (m^2^/g)**	**Micropore Volume (cm^3^/g)**	**Mesopore Volume (cm^3^/g)**	**Micropore Vol. Percent (%)**
Graphite	140.0	−0.38	6.7	0.002	0.040	4
YP50F	1.7	1471.35	204.8	0.586	0.191	75
YP80F	1.7	1345.30	1029.5	0.571	0.656	47
Graphite-LiOH	76.49	0.11	15.11	0.004	0.090	4
YP50F-LiOH	76.5	−3.28	15.0	0.003	0.072	3
YP80F-LiOH	1.7	1238.64	917.1	0.526	0.578	48
**T-Plot**						
**Samples**	**Total Surface Area (m^2^/g)**	**Micropore Surface Area (m^2^/g)**	**External Surface (m^2^/g)**	**Micropore Volume (cm^3^/g)**	**Mesopore Volume (cm^3^/g)**	**Micropore Vol. Percent (%)**
Graphite	7.40	2.44	23.96	−0.021	-	
YP50F	1898.4	1639.4	36.7	0.693	0.084	89
YP80F	2226.9	2304.4	70.4	1.084	0.100	88
Graphite-LiOH	1.665	2.30	21.283	0.023	-	
YP50F-LiOH	43.5	11.59	11.61	0.037	0.111	50
YP80F-LiOH	1063.2	1094.1	46.0	0.511	0.143	84

**Table 3 molecules-26-02187-t003:** The area, specific capacitance, energy density and power density calculation of each electrode from CV test.

№	Sample	Current Collector	Area of Integration	Specific Capacitance (Cp) (F/g)	Energy Density (Wh/kg)	Power Density (kW/kg)
1	Li-YP80F	Cu foil	9.828 × 10^−4^	1.196 × 10^−4^	14.95	66.44
		Ni foil	3.918 × 10^−4^	2.118 × 10^−5^	2.65	13.81
2	Li-YP50F	Cu foil	7.601 × 10^−4^	1.169 × 10^−4^	14.61	59.77
		Ni foil	2.21 × 10^−4^	1.21 × 10^−5^	1.51	7.26
3	Li-Graphite	Cu foil	6.559 × 10^−4^	1.17 × 10^−4^	14.62	57.83
		Ni foil	2.158 × 10^−4^	1.192 × 10^−5^	1.49	6.79

## Data Availability

Data can be made available upon written request to the corresponding author and with a proper justification.

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
