# Peer review of "Surface Modification Effect and Electrochemical Performance of LiOH-High Surface Activated Carbon as a Cathode Material in EDLC"

_molecules, 2021, doi:10.3390/molecules26082187_

Round 1
Reviewer 1 Report
This work is a research aimed at optimizing activated carbon interface. After reading this article, I do not recommend publishing this manuscript in its current form. The decision given is to reject the manuscript. The specific reasons are as follows:
- The key significance of this work is to increase the surface functional groups of activated carbon, promote the interface wettability and other physical and chemical properties, then improve the application performance in electric double layer capacitors. However, the article does not give a direct wettability test to prove the improvement of the material after modification.
- The overall description of the article lacks logic, the language is not concise and difficult to understand. For example, an overview of the experiment process should add at the beginning of the results, so that readers have an overall understanding of the research.
- There are few data in the article, and only 10 cycles of data in the application part are not enough to support the conclusion. In addition, , Figure (a) in Figure 5 already contains the test results of 3 materials, (b-d) do not need to list again.
- There are many basic errors in the article editing. The abbreviation for milliliter should be “mL” (line 316 and 318). The font size in introduction part is not uniform. The abbreviations for hours are not uniform (line 291 and 297).
- Figure 11 does not directly reflect the electrochemical advantages of YP80F-LiOH
- The expression abbreviation is not standardized. The abbreviation of the working electrode (WE) is mentioned in line 332, but it has been used before.
In summary, the article quality needs to be greatly improved and the experimental data should be more adequate.
Author Response
Reviewer 1
This work is a research aimed at optimizing activated carbon interface. After reading this article, I do not recommend publishing this manuscript in its current form. The decision given is to reject the manuscript. The specific reasons are as follows:
- The key significance of this work is to increase the surface functional groups of activated carbon, promote the interface wettability and other physical and chemical properties, then improve the application performance in electric double layer capacitors. However, the article does not give a direct wettability test to prove the improvement of the material after modification.
- Thank you for your comment. I checked wettability properties of the LiOH-treated YP80F, YP50F and Graphite, and pure material.
“The wettability of the LiOH-treated carbon materials was tested by gravimetric methods using water and 5% KOH. The measurement-based of determining how much amount of solvent (water, 5% KOH) absorbed in the working electrode which using a LiOH-treated carbon material and pure carbon material. The results of the wettability properties of all samples are summarized in Bit-graph. According from the summarized result, LiOH-treated sample has high wettability properties, due to the acidic groups of the surface which can affect the impregnation of the electrolyte. The main purpose of the study was to change the surface of the carbon material by the method of functional groups with hydrophilic and hydrophobic properties, depending on the type of electrolyte applied. The results of the experiments confirm that the surface of the carbon material has been successfully modified to a hydrophilic quality using LiOH-oxidizing agents.”
Bit-graph. The wettability of the carbon samples.
- The overall description of the article lacks logic, the language is not concise and difficult to understand. For example, an overview of the experiment process should add at the beginning of the results, so that readers have an overall understanding of the research.
- Thank you for your comment. The result and discussion part of the main manuscript was revised, and new added part remarked in the manuscript.
- “The XRD pattern of LiOH-treated high surface area of activated carbon after heat treatment at 600°C is shown in Figure 1.”
- “The study of how the surface and functional group of a carbon material changes after LiOH treatment and calcification will determine how the material interacts with alkaline electrolyte ions. The surface functional group and surface area with pore distribution of the samples were analyzed by a Boehm titration and N2-adsorption-desoprtion isotherm. Boehm titration is an acid-base titration method that is used to determine the number of surface oxygen groups (acidic or basic) present on LiOH-treated carbon surfaces. This method is usually used complementary with other methods such as Fourier Transform Infrared Spectroscopy (FTIR).”
- “The high surface area of YP50F, YP80F activated carbon and graphite was modified with LiOH-oxidizing agent and heat treatment (600 °C). HSAC materials with a micro-porous structure can be synthesized by (LiOH) oxidizing agent treatment for the high surface area of activated carbon under the high temperature. In addition, after the activated carbon reacting with oxidizing agents, the number of basic functional groups decreases, the number of hydrophilic groups increases, or the hydrophilicity of the activated carbon surface improves, which can improve the long-term reliability of the cell. The Li-HSAC samples were used for the working electrode by following a Doctor blade” method.
- “The use of Li-YP80F, a material with a high surface area, as a working electrode allows for more charge exchange and, on the other hand, allows more electrons to accumulate over a given period of time and conduct more current. The charge exchange happens at a fixed rate per unit area at the surface of the electrode.”
- There are few data in the article, and only 10 cycles of data in the application part are not enough to support the conclusion. In addition, , Figure (a) in Figure 5 already contains the test results of 3 materials, (b-d) do not need to list again.
- Thank you for your comment. Figure 5 revised and new cyclic performance was added in the main manuscript.
“The CV test was run for 10 and 100-cycles with 100 mV/sec scan rate. The CV graph and current value were different in each sample according to the working electrode. The CV result of Li-YP80F is shown in Figure 8 (a) (b). The anodic and cathodic peak current value and peak segregation of Li-YP80F were better than those of the other two samples, due to the micropore-size and physical properties of the sample. The oxidation peak current value appeared at 0.0055 mA/cm2 current density and the reduction peak value at -0.0014 mA/cm2, when the Li-YP80F sample used to the Cu-foil surface. The redox peaks appeared at 0.0025 mA/cm2 and -0.0009 mA/cm2, in the case of using a Nickel foil, as shown in Figure 8 (b).
As shown in Figure 8 (c), the CV curve had one sharp peak in the oxidation state. The oxidation peak appeared at 0.0035 mA/cm2 and the reduction-peak at the 0.00042 mA/cm2 regions. When the Li-YP50F covered on the nickel foil, the redox peaks appeared at 0.00231 mA/cm2 and -0.00051 mA/cm2 current density regions, as shown in Figure 8 (d). For the Li-graphite, the current density of the oxidation peak appeared at 0.0053 mA/cm2 and the reduction peak at (-0.00045 mA/cm2), as shown in Figure 8 (e). In the case of Li-graphite coated on the nickel foil surface, the oxidation peak appeared at 0.0013 mA/cm2 and reduction peak appeared at -0.00083 mA/cm2 region, as shown in Figure 8 (f).
Figure 8. Cyclic voltammetry (CV) test of LiOH-treated YP80F, YP50F and Graphite (Current collector -Cu foil and Ni foil and 5% KOH electrolyte) under 10 cycles.
In addition, the Cu plate collector has a conductivity of 0.6 S/cm and it can support the activity of the base sample. The electrochemical stability of cathode materials was tested by 100 recycling tests, as shown in Figure 9. The peak-to-peak separation and high redox peaks appeared in the Cu-foil used three electrode system. After 100 recycling tests, peak current drop decreased the peak profile became stable. The Cu-current collector of the working electrode was corroded, and the surface of the electrode did not change after the electrochemical test. In the case of nickel foil, there is no corrosion process appeared and the current peak did not significantly change. The corrosion process can be happened due to the electrolyte and concentration.”
Figure 9. Cyclic voltammetry (CV) test of LiOH-treated YP80F, YP50F and Graphite (Current collector -Cu foil and Ni foil and 5% KOH electrolyte) under 100 cycles.
Figure 10 shows the current value of the working electrode (vs. Ag/AgCl) according to time-condition in 5% KOH electrolyte. The current of the working electrode was high in the first cycle and decreased to a certain extent after 10 cycles. The profile of Li-YP80F had a uniform structure and this result confirmed that Li-YP80F sample had more stable structure and the working performance.
Figure 10. Stability test of LiOH-treated YP80F, YP50F and Graphite (Cu-foil and Ni-foil current collector and 5% KOH electrolyte).
- There are many basic errors in the article editing. The abbreviation for milliliter should be “mL” (line 316 and 318). The font size in introduction part is not uniform. The abbreviations for hours are not uniform (line 291 and 297).
- Thank you for your comment. The abbreviation of the milliliter and hour was revised in the main manuscript and the revised part highlighted as yellow. The font size of the introduction part and paragraph spacing were revised.
- Figure 11 does not directly reflect the electrochemical advantages of YP80F-LiOH
- Thank you for your comment. I revised Figure 11 and re-attached it in the main manuscript. In addition, I added more figures in the main manuscript and the numberings of Figure 11 changed into Figure 13.
Figure 13. Schematic illustration of the preparation of LiOH-treated carbon material and electrochemical analysis.
- The expression abbreviation is not standardized. The abbreviation of the working electrode (WE) is mentioned in line 332, but it has been used before.
- Thank you for your comment. The word “working electrode” did not abbreviated in the main manuscript and the words were highlighted as a yellow color.
In summary, the article quality needs to be greatly improved and the experimental data should be more adequate.
Reviewer 2 Report
The present work aims to show the performance of the activated carbon-based cathode for electrochemical double layer capacitor. It is a very hot topic since the goals of overall researchers in this field are the development of EDLC with high energy density. The authors present interesting results, but they did not demonstrate the electrochemical stability of cathode materials. At least 100 cycles are needed before to claim (line 224) “The profile of Li-YP80F had a uniformly reduced structure and this result confirmed the structure’s stability and the material’s performance.”
Additional comments:
- Please define EDLC
- (abstract) the performances of as-prepared materials are missing.
- Line 28: it should be “cathode material for EDLC” instead of “half-cell cathode electrode”
- Generally, the performance of capacitor electrode is expressed as specific capacitance (in F/g), as energy density (Wh/kg) at given powder density (W/g). These quantities are missing in this work.
- Line 46: define AC
- (introduction) the aim of the work is missing.
- It should be interesting to probe the surface state of lithiated samples using Raman spectroscopy, which is also a good tool to probe the quality of carbon (degree of graphitization).
- Line 165: the authors stated “The micropore-structured carbon material showed a high capacitance” but did not quantify the capacitance.
- What is the carbon particle size?
- (Table 2) There is a big difference between mesopore diameters. Why? Data of Table 2 are poorly discussed.
- (section 2.3) The authors should discuss the redox potentials and compare their results with literature data.
- (Fig.7a) the CV profile is segmented. Why?
- 8 displays a fast decay of peak current? Why? It seems that theses systems are not electrochemically stable. What is the electrode surface state after 10 cycles?
- Line 220; unit of conductivity is Siemens per centimeter (S/cm)
- (experimental methods) please define
-the XRD wavelength
- the FTIR spectral resolution
- the composition of the electrolyte
- the cathode current collector
In conclusion, this manuscript needs major revision with additional experiments.
Author Response
Reviewer 2
The present work aims to show the performance of the activated carbon-based cathode for electrochemical double layer capacitor. It is a very hot topic since the goals of overall researchers in this field are the development of EDLC with high energy density. The authors present interesting results, but they did not demonstrate the electrochemical stability of cathode materials. At least 100 cycles are needed before to claim (line 224) “The profile of Li-YP80F had a uniformly reduced structure and this result confirmed the structure’s stability and the material’s performance.”
Additional comments:
- Please define EDLC
- Thank you for your comment. Electric double layer capacitor(EDLC) is the electric energy storage system based on charge–discharge process (electro sorption) in an electric double layer on porous electrodes, which are used as memory back-up devices because of their high cycle efficiencies and their long life-cycles. EDLC supercapacitor construction is similar to a battery in that there are two electrodes immersed in an electrolyte, with an ion-permeable separator located between the electrodes to prevent electrical contact. The porous structure with a high surface of carbon material is an ideal material for the electrode on the EDLC.
The EDLC has several advantages, and all are summarized as follows: (1) possible to charge/discharge quickly, (2) long cycle life because of no chemical reactions, just adsorption/desorption of electrolyte ions, (3) high efficiency for charge/discharge cycle, (4) no heavy metals used and consequently environment friendly, and (5) possible to discharge in high current density.
- (abstract) the performances of as-prepared materials are missing.
- Thank you for your comment. The discussion of the LiOH-treated carbon materials was added in the abstract and the part remarked as a yellow-color.
“This study aimed to improve the performance of the activated carbon-based cathode by increasing the Li content and to analyze the effect of the combination of carbon and oxidizing agent. LiOH is an alkaline hydroxide that has high reactivity with carbon materials. The high surface area of activated carbon supports the diffusion of a large amount of Li ions, which is beneficial for improving the electrochemical properties. The hydrothermal method and heat-treatment process were used to synthesize the Li-high surface area activated carbon material (Li-HSAC). The crystal structure of Li-HSAC was analyzed by X-ray diffraction (XRD), the surface state and quantitative element by scanning electron microscopy with energy dispersive X-ray spectroscopy (SEM-EDX) and the surface properties with pore-size distribution by Brunauer–Emmett–Teller (BET), Barrett–Joyner–Halenda (BJH) and t-plot methods. The specific surface area of the Li-YP80F is 1063.2 m2/g, micropore volume value is 0.511 cm3/g and mesopore volume is 0.143 cm3/g, and these all values are higher than other LiOH-treated carbon. These results demonstrated that Li-YP80F has a more active interfacial site between the alkaline electrolyte and the cathode. The surface functional group was analyzed by a Boehm titration, and the higher number of acidic groups compared to the target facilitated the improved electrolyte permeability, reduced the interface resistance and increased the electrochemical properties of the cathode. The oxidizing agent of LiOH treated high surface area of activated carbon was used for the cathode material for EDLC to determine its electrochemical properties and the as-prepared electrode retained excellent performance after 10 cycles and 100 cycles. The electrochemical results confirmed that oxidizing agent treatment with calcination can enhance the performance of the high surface area of carbon materials. The anodic and cathodic peak current value and peak segregation of Li-YP80F were better than those of the other two samples, due to the micropore-size and physical properties of the sample. The oxidation peak current value appeared at 0.0055 mA/cm2 current density and the reduction peak value at -0.0014 mA/cm2, when the Li-YP80F sample used to the Cu-foil surface. The redox peaks appeared at 0.0025 mA/cm2 and -0.0009 mA/cm2, in the case of using a Nickel foil, after 10 cycling test. The electrochemical stability of cathode materials was tested by 100 recycling tests. The peak-to-peak separation and high redox peaks appeared in the Cu-foil used three electrode system. After 100 recycling tests, peak current drop decreased the peak profile became stable. The LiOH-treated high surface area of activated carbon had synergistically upgraded electrochemical activity and superior cycling stability that were demonstrated in EDLC.”
- Line 28: it should be “cathode material for EDLC” instead of “half-cell cathode electrode”
- Thank you for your comment. The words are changed, and revised part highlighted as a yellow-color in the main manuscript.
- Generally, the performance of capacitor electrode is expressed as specific capacitance (in F/g), as energy density (Wh/kg) at given powder density (W/g). These quantities are missing in this work.
- Thank you for your comment. We calculated each quantities by following the below equations.
“The specific capacitance, area of the integration, energy density (Wh/kg) at given powder density (kW/kg) are important factors which define the performance of the working electrode. Aforementioned mentioned all factors were calculated by following the below equations. All calculated results are summarized in Table 3. The calculation of specific capacitance needs to summarize some units, such as charge, capacitance.”
The calculation of specific capacitance needs to summarize some units, such as charge, capacitance.
- Combined equation was the Cp
The calculation of energy density (Wh/kg) and powder density (W/g).
Table 3. The area, specific capacitance, energy density and power density calculation of each electrode from CV test.
â„– |
Sample |
Current collector |
Area of Integration |
Specific capacitance (Cp) (F/g) |
Energy density (Wh/kg) |
Power density (kW/kg) |
1 |
Li-YP80F |
Cu foil |
9.828x10-4 |
1.196x10-4 |
14.95 |
66.44 |
Ni foil |
3.918x10-4 |
2.118x10-5 |
2.65 |
13.81 |
||
2 |
Li-YP50F |
Cu foil |
7.601x10-4 |
1.169x10-4 |
14.61 |
59.77 |
Ni foil |
2.21x10-4 |
1.21x10-5 |
1.51 |
7.26 |
||
3 |
Li-Graphite |
Cu foil |
6.559x10-4 |
1.17x10-4 |
14.62 |
57.83 |
Ni foil |
2.158x10-4 |
1.192x10-5 |
1.49 |
6.79 |
- Line 46: define AC
- Thank you for your comment. The definition of the AC was added in the main manuscript and remarked as a yellow-color.
- (introduction) the aim of the work is missing.
- Thank you for your comment. The revised part added in the main manuscript and remarked as a yellow-color.
“The following sub-groups are important goals of this research: (a) surface hydrophilicity control technology improves the output characteristics by improving the electrolyte impregnation at the interface of the electrode and activated carbon and reducing the interface resistance, (b) improves the long-term reliability of the cell by inhibiting the corrosion of the hydrogen storage material of the cathode through the hydrophilicity control technology and catalytic function technology of the activated carbon surface, and (c) batch-type hydrothermal synthesis and microwave control technology for mass production. In this study, LiOH- treated HSACs were successfully synthesized with surface modification that can activate the ion transfer and electrolyte impregnation. The phase structure, surface morphology, element analysis, functional groups, and electro-chemical properties of the modified (including surface coating and surface structure modification) activated carbon were analysed using XRD, SEM, EDX, FTIR, BET and Potentiastat apparatus. LiOH-treated carbon material used for the working electrode. As-prepared Li-YP80F sample proved to have exceptional electrochemical performance, with high pore distribution and specific surface area. A LiOH-treated HSAC had good stability under a 10 and 100-cycling test. The peak current density and peak separation profile was good in Li-YP80F sample. A Li-YP80F electrode had good stability under a 100-cycle test. By comparing these three cycling performance curves, it can be seen that there is a process of activating capacity that happens during the first cycles. This approach presented herein offers a promising route for the rational design of a new class of supercapacitors.”
- It should be interesting to probe the surface state of lithiated samples using Raman spectroscopy, which is also a good tool to probe the quality of carbon (degree of graphitization).
- Thank you for your comment. We added the Raman spectra of the LiOH-treated all carbon samples.
Figure 7. Raman spectra of the LiOH-Graphite, LiOH-YP50F and LiOH-YP80F sample.
“ Figure 7 expresses the Raman spectra of the LiOH-treated graphite, YP50F and YP80F samples. Raman spectroscopy is the chemical-analysis method which can analyze the chemical structure phase and molecular interaction of matter. The principle of Raman spectroscopy is a light-scattering technique, where the light interact to the molecule bond and the result provides the information of carbon material (D and G-band). The assignment of the D and G peaks is straightforward in the “molecular” picture of carbon materials. The Raman spectrum of LiOH-YP50F and LiOH-YP80F is dominated by two prominent features. The G band near 1,597 cm-1 and 1591 cm-1 is due to the degenerate zone-center (G) optical phonon mode with E2g symmetry, which is found in most graphitic materials. The most important is the D band near 1,332 cm-1 and 1,338 cm-1 due to a one-phonon scattering involving a phonon near the K point and scattering due to defects [27].
Two broad peaks at 2657, 2662, 2914 and 2904 cm-1 observed in the spectrum of LiOH was also detected in the Raman spectrum of LiOH-treated YP50F and YP80F. These two peaks are assigned to the asymmetric (Bg) and symmetric (Ag) stretches [28].
The five peaks are dominated in LiOH-Graphite sample. The strong band at 2,693 cm-1 is due to a two-phonon scattering involving phonons near the K point and is called the 2D band. The 2D band is usually stronger than the G* and G' band although it is due to a two-phonon scattering process. The triple resonance process, where the electron and the hole scatter simultaneously, can also explain the two-phonon scattering. The characteristic peak of LiOH not obtained in LiOH-Graphite sample, due to the peak intensity of each components.”
- Line 165: the authors stated “The micropore-structured carbon material showed a high capacitance” but did not quantify the capacitance.
- Thank you for your comment. We calculated the specific capacitance of the LiOH-treated graphite, YP50F and YP80F carbon. All analysis results are summarized in Table 3. According to the analysis result, Li-YP80F sample had high value of specific capacitance and integration area. This sample had high number of micropore distribution and high surface area with hydrophilicity, and the porous (micro) structure with a high surface of carbon material is an ideal material for the electrode on the EDLC. Aforementioned all factors have in Li-YP80F, it can define the electrochemical properties of the material.
- What is the carbon particle size?
- Thank you for your comment. The particle size of the natural graphite is <45 μm.
The particle sizes of the YP50F and YP80F are 1-2 μm. According to SEM images, the particle size of the LiOH-graphite is 13 μm, LiOH-YP50F has 8 μm, LiOH-YP80F has 9 μm.
- (Table 2) There is a big difference between mesopore diameters. Why? Data of Table 2 are poorly discussed.
- Thank you for your comment. The results of Li-HSAC pore-volume and specific surface area are summarized in Table 2.
“HSAC with large micropore structure may be obtained due to the LiOH agent and high temperature calcination. From BET analysis, the total pore volume and mean pore diameter of raw carbon material is reduced due to the oxidizing agent treatment. LiOH-oxidizing agent gathered in the disordered pore structure of the YP50F and YP80F. In the case of the graphite, the LiOH located on the surface. The pore distribution of the LiOH-treated samples was analyzed by BJH and T-plot method. According from the result, mesopore and micropore volumes and pore surface area were decreased. The micropore volume and surface area of the YP80F was high, the high amount of LiOH-oxidizing agent can inter-collected in the pore. The inter-collected amount of the LiOH can positively improve the Li+ ion transition between cathode and anode of EDLC, then it can upgrade the performance of the EDLC material”
- (section 2.3) The authors should discuss the redox potentials and compare their results with literature data.
- Thank you for your comment. Redox potential is a measure of the tendency of a chemical species to acquire electrons from or lose electrons to an electrode and thereby be reduced or oxidized, respectively. The redox potential of the Li-YP80F is 0.30V (Cu-foil) and 0.21V (Ni-foil), the potential of the Li-YP50F is 0.114V (Cu-foil) and 0.09V (Ni-foil). In the case of the Li-Graphite, the redox potential is 0.182V (Cu-foil) and 0.176V (Ni-foil). According to the results, the redox potential of the Li-YP80F is high due to the low resistivity between the electrode surface and electrolyte, pore-distribution and hydrophilic capabilities of the working electrode.
- (Fig.7a) the CV profile is segmented. Why?
- Thank you for your comment. The all-cyclic voltammetry graph was segmented, it determines the oxidation and reduction peak current-density for each cycle. In addition, it is clearly indicated how the charge density changes with each cycle.
- 8 displays a fast decay of peak current? Why? It seems that these systems are not electrochemically stable. What is the electrode surface state after 10 cycles?
- Thank you for your comment. The concentration of the electrolyte can affect to the performance of the working electrode. Mainly, the high-concentration of electrolyte usually corrosion at the electrode surface or current collector, which affect the performance. In the case of LiOH-graphite, the peak current was significantly reduced because it has low stability properties. LiOH-treated YP80F and YP50F activated carbon has better activity and stability. The Cu-current collectors of the working electrode were corroded. The surface of the electrode did not change after the electrochemical test.
- Line 220; unit of conductivity is Siemens per centimeter (S/cm)
- Thank you for your comment. The unit of the conductivity was changed and highlighted as a yellow color.
- (experimental methods) please define.
- Thank you for your comment. The experimental conditions are added in the main manuscript and written in below part.
-the XRD wavelength
- the FTIR spectral resolution
- the composition of the electrolyte
- the cathode current collector
In conclusion, this manuscript needs major revision with additional experiments.
- The crystal phases of the samples samples were examined between 2θ = 10 to 70° at a scan rate of 1° min-1using an X-ray diffraction instrument (SHIMADZU XRD-6000) equipped with a Cu Ka X-ray source (1.5406 Å).
- The functional group and chemical bonds were analyzed by using a Fourier-transform infrared spectrometer (FTIR iS5, Thermoscience), and the spectral resolution was 3.8 cm-1 background scanning speed and sample scanning speed are 20 scans, respectively.
- The typical three-electrode assembly was immersed in 5% KOH supporting electrolyte solutions. KOH is a basic electrolytes and it has been the most extensively used because of its high ionic conductivity and it can support the cyclic stability of the working electrode.
- The working electrode (WE) was prepared by following a “Doctor blade” method. Ethyl cellulose was used as a binding material and mixed with Li-HSAC in a 1: 3 ratio. Then, a few drops of pure ethanol were added, and the resulting mixture was ground and used to veneer the copper foil-top. The size of the copper foil and nickel foil is 2×2 cm. The Cu-foil current collector enhanced the capacity of the sample; this is due to the conductivity of copper being about 6 S/cm. The uniformity and thickness were metered by adjusting gap between the blade and the substrate. The main principal process of doctor-blading uses a frame with a reservoir coating liquid which is moving relatively to the substrate. When a constant movement between the blade and substrate, the semiliquid mixture spread onto the substrate to make a thin thin-sheet and after drying process it can turn into a gel-layer.
Round 2
Reviewer 1 Report
The revised manuscript is well improved and the comments are well-addressed. But the abstract part of the revised manuscript is too long, it is recommended to modify it more concisely.
Author Response
We revised abstract part, comprehensively.
Reviewer 2 Report
After an accurate revision, this paper can be published as it is.
Author Response
Thank you for your valuable comments.